# Design, Synthesis and Biological Investigation of Flavone Derivatives as Potential Multi-Receptor Atypical Antipsychotics

**DOI:** 10.3390/molecules25184107

**Published:** 2020-09-08

**Authors:** Lanchang Gao, Zhengge Yang, Jiaying Xiong, Chao Hao, Ru Ma, Xin Liu, Bi-Feng Liu, Jian Jin, Guisen Zhang, Yin Chen

**Affiliations:** 1Department of Biomedical Engineering, College of Life Science and Technology, Huazhong University of Science and Technology, Wuhan 430074, China; D201577449@hust.edu.cn (L.G.); charles@long-ou.com (Z.Y.); bear@hust.edu.cn (J.X.); d201880507@hust.edu.cn (C.H.); xliu@mail.hust.edu.cn (X.L.); bfliu@mail.hust.edu.cn (B.-F.L.); 2Jiangsu Key Laboratory of Marine Biological Resources and Environment, Jiangsu Key Laboratory of Marine Pharmaceutical Compound Screening, School of Pharmacy, Jiangsu Ocean University, Lianyungang 222005, China; 12111030011@fudan.edu.cn (R.M.); 2019000016@jou.edu.cn (J.J.)

**Keywords:** dopamine, serotonin, multi-target, atypical antipsychotics

## Abstract

The design of a series of novel flavone derivatives was synthesized as potential broad-spectrum antipsychotics by using multi-receptor affinity strategy between dopamine receptors and serotonin receptors. Among them, 7-(4-(4-(6-fluorobenzo[d]isoxazol-3-yl) piperidin- 1-yl) butoxy)-2,2-dimethylchroman-4-one (**6j**) exhibited a promising preclinical profile. Compound **6j** not only showed high affinity for dopamine D_2_, D_3_, and serotonin 5-HT_1A_, 5-HT_2A_ receptors, but was also endowed with low to moderate activities on 5-HT_2C_, α_1_, and H_1_ receptors, indicating a low liability to induce side effects such as weight gain, orthostatic hypotension and QT prolongation. In vivo behavioral studies suggested that **6j** has favorable effects in alleviating the schizophrenia-like symptoms without causing catalepsy. Taken together, compound **6j** has the potential to be further developed as a novel atypical antipsychotic.

## 1. Introduction

Schizophrenia is a severe neuropsychiatric disorder that affects over 1% of the world’s population [1]. The clinical symptoms of schizophrenia are categorized into three general types, including positive, negative, and cognitive symptoms [2]. Currently, antipsychotics are the mainstay in schizophrenia treatment. For example, the first-generation antipsychotics (FGAs), namely typical antipsychotics including chlorpromazine and haloperidol, are all dopamine D_2_ receptor antagonists and effective for the treatment of positive symptoms, but invalid for the negative symptoms and cognitive impairment [3,4]. However, the strong antagonism of D_2_ receptor results in serious side effects such as extrapyramidal symptoms (EPS), hyperprolactinemia, and cognitive impairments [5]. The second-generation antipsychotics (SGAs), also known as atypical antipsychotics, including clozapine [6], and risperidone [7], have antagonistic action on both dopamine and serotonin receptors, especially with high affinity for 5-HT_1A_ and 5-HT_2A_ receptors, which contributes to various kinds of therapeutic advantages, such as fewer EPSs than FGAs [8]. However, the SGAs also produce many adverse metabolic and cardiovascular effects [9], such as weight gain, hyperprolactinemia, constipation, QT prolongation, and also show limited effects on negative symptoms and cognitive impairments [10,11]. Fortunately, these defects could be avoided by developing a multi-receptor ligand with a “selective” multi-receptor profile which can precisely modulates several specific targets to get a better effect and safety in the therapy of schizophrenia [12,13].

To verify this multi-receptor affinity strategy, we have previously developed several novel potential antipsychotics, such as compounds **1** and **2** shown in Figure 1, both of which could significantly alleviate the positive symptoms of schizophrenia, are associated with lower weight gain and lower prolactin levels, and have obvious therapeutic effects. In order to obtain versatile molecules with better therapeutic effects and less side effects, in this paper, a series of new compounds was designed and synthesized by means of a polypharmacological strategy and molecular hybridization method on the basis of our previous research [14,15]. The design concept of new compounds as shown in Figure 1, where the general structure contains a flavone-like moiety, a heterocyclic or arylpiperazine (piperidine) moiety and a flexible linker. By optimizing the combination of the benzocaprolactam fragment of brexpiprazole and the coumarin derivative scaffolds of compounds **1** and **2**, they were transformed into a flavone-like fragment. The choice of heterocyclic or arylpiperazine (piperidine) originates from the commercial atypical antipsychotics [16], such as risperidone, brexpiprazole, aripiprazole. The linker flexibility of the new compounds also be evaluated as our previous SAR studies have proved it to play an important role in modulating the receptor function profiles [14,15,17].

Among these derivatives, compound **6j** [7-(4-(4-(6-fluorobenzo[d]isoxazol-3-yl) iperidin-1-yl) butoxy)-2,2-dimethylchroman-4-one] exhibited a favorable polypharmacological antipsychotic profile. In vitro, it showed much higher potency for the desired targets (D_2_, 5-HT_1A_, 5-HT_2A_, and D_3_) than other off-target receptors (5-HT_2C_, *α*_1_, and H_1_). Further in vivo behavioral studies suggested that it has favorable effects in alleviating schizophrenia-like symptoms without causing catalepsy and a low liability to induce side-effects. Thus, compound **6j** has the potential to be developed as a novel atypical antipsychotic candidate to treat schizophrenia.

## 2. Results and Discussion

### 2.1. *Chemistry*

New compounds were synthesized as shown in Scheme 1, Scheme 2 and Scheme 3. In Scheme 1, 2,4-dihydroxyacetophenone (**3**) was converted into intermediate **4** by reaction with acetone in the presence of pyrrolidine and acetonitrile, and the intermediates **7a** and **7b** were prepared using a similar process. Compounds **5a**–**5b** and **8a**-**8d** were obtained through alkylation of **4** and **7a**–**7b** with 1-bromo-3-chloropropane or 1-bromo-4-chlorobutane. Compounds **6a**–**6j** and **9a**–**9k** were obtained by coupling **5a**–**5b** and **8a**-**8d** with different arylpiperazines or heterocyclic arylpiperidines in the presence of K_2_CO_3_ in acetonitrile. As shown in Scheme 2, the new compounds **11a**–**11e** and **14a**–**14b** were synthesized following similar approaches to those described in Scheme 1, that is treatment of resorcinol and its derivatives **10a**-**10e** with trifluoromethanesulfonic acid in the presence of 3-chloropropionic acid, and then treatment with sodium hydroxide solution to afford **11a**-**11e**. Compounds **14a** and **14b** were synthesized starting from resorcinol and its derivatives by treatment with crotonic acid in the presence of anhydrous zinc chloride. The preparation of **12a**–**12e** and **15a**–**15b** used the same procedure as the synthesis of **5a**-**5b**. Subsequently the intermediates **12a**–**12e** and **14a**–**14b** were coupled with 6-fluoro-3-(piperidin-4-yl) benzo[d]isoxazole to yield compounds **13a**-**13d** and **16a**–**16b**. Compound **21** was synthesized in five steps as described in Scheme 3. The intermediate **17** was synthesized from 2,4-dihydroxypropiophenone. Hydrolysis of **17** under basic conditions (NaHCO_3_/MeOH) produced **18**. Intermediate **19** was prepared by reduction with Pd/C, followed by reaction with 1-bromo-4-chlorobutane to afford **20**. The target compound **21** was obtained by coupling **20** with 6-fluoro-3-(piperidin-4-yl)benzo[d]isoxazole in the presence of K_2_CO_3_ in acetonitrile.

### 2.2. In Vitro Evaluation and Structure-Activity Relationship (SAR)

At beginning of this study, the 7-hydroxy-2,2-dimethylchroman-4-one group was selected as a privileged structure derived from coumarin and flavone. The influence of various heterocyclic and aryl piperazines(piperidine) on the activities of D_2_, 5-HT_1A_ and 5-HT_2A_ receptors were primarily evaluated and the results are summarized in Table 1. Among those compounds, **6a**–**6d** bearing phenylpiperazine and substituted phenylpiperazines (-F, -CF_3_, -OCH_3_) showed moderate affinity for 5-HT_1A_ and 5-HT_2A_ receptors and lower affinity for D_2_ receptors. When the phenylpiperazine was replaced with a heterocyclic piperazine, compound **6e** bearing a pyripiperazine devoid of affinity for D_2_ receptor resulted. Compound **6f** with pyrimidine almost lost the affinity for all the three receptors. Compound **6g** bearing a 1-(benzo[d][1,3] dioxol-5-ylmethyl) piperazine showed lower affinity for the D_2_ receptor. Interestingly, replacing the phenylpiperazine with a benzoheterocycle-piperazines(piperidine), such as benzothiazole-piperazine and benzisothiazole- piperazine, gave compounds **6h** and **6i**, which showed significantly increased for all three receptors. Especially, compound **6j** bearing a 6-fluorobenzo[d]isoxazole-piperidine moiety displayed excellent D_2_ (*K*_i_ = 8.1 nM), 5-HT_1A_ (*K*_i_ = 9.7 nM) and 5-HT_2A_ (*K*_i_ = 3.2 nM) receptor potency.

According to the above results, the 6-fluoro-3-(piperidin-4-yl)benzo[d]isoxazole moiety was selected as a privileged fragment, as our previous studies have proved that compounds bearing this pharmacophore show favorable activities for D_2_, 5-HT_1A_ and 5-HT_2A_ receptors [14,15,17]. Next the effect of substituents on the 7-hydroxy-2,2-dimethylchroman-4-one group (Table 2) was investigated, such as five-membered rings, six-membered rings, methyl and chlorine groups. First, when dimethyl was replaced with a cyclopentyl ring, the obtained compound **9d** had reduce activities on the D_2_, 5-HT_1A_ and 5-HT_2A_ receptors (**9d** vs. **6j**). Similarly, enlargement of the five-membered ring to a cyclohexyl ring (compound **9h**) slightly lowered the affinity for the three receptors compared with **6j**. Removing the dimethyl or replacing the dimethyl with a methyl (**13a** and **16a**) showed a negative impact on all the three receptors compared with compound **6j**. Meanwhile, when the two methyl substituents at the 5 and 6 positions (compounds **21**) of 7-hydroxy-2,2-dimethylchroman-4-one group were changed simultaneously, it resulted in increased activities for the 5-HT_2A_ receptor but decreased activity for the D_2_ and 5-HT_1A_ receptors. We further investigated the influence of a methyl group substituted at different positions on the phenyl ring. The activities of the substitution of methyl at the 2- (**13c**), 3- (**13b**) and 8- (**13d**) positions of the phenyl ring were retained, but were slightly lower than without a substituent (**13a**). According the results, when the methyl was substituted at the 6 position of 7-hydroxy-2,2-dimethylchroman-4-one group (**16a)** and the 8-chloro substituted derivative **16b** displayed decreased activity for the 5-HT_1A_ receptor (**16a** vs. **16b**).

As the compounds bearing a cyclopentyl ring (**9c** and **9d**) and a cyclohexyl ring (**9g** and **9h**) moiety showed higher potency for the three receptors (Table 1 and Table 2), we therefore attempted to expand the structural transformation of 6-fluoro-3-(piperidin-4-yl) benzo[d]isoxazole fragment. When the privileged structure 6-fluoro-3-(piperidin-4-yl) benzo[d]isoxazole was replaced with phenylpiperazine derivatives and heterocyclic piperazines, such as 1-(2-methoxyphenyl) piperazine (**9a**,**9b**,**9e**),1-(2-(trifluoromethyl)phenyl)piperazine (**9f**), 1-(2,3-dimethylphenyl)piperazine (**9i**), 1-(2,3-dichlorophenyl)piperazine (**9j**) and 3-(piperidin-4-yl)benzo[d]isothiazole (**9k**), all of the resulting compounds fail to improve the affinities for the D_2_, 5-HT_1A_ and 5-HT_2A_ receptors, indicating that the 6-fluoro-3-(piperidin-4-yl)benzo[d]isoxazole was the optimum structure for this type of compound.

Our previous SAR studies reveal that the linker length and flexibility play an important role in the regulation of the receptor potency, and the lengths of the linker generally considered appropriate were three carbons and four carbons 1415. Therefore, investigation of chain length was focused on three carbons and four carbons. As shown in Table 1 and Table 2, most of the compounds with chain lengths of four carbons exhibited higher affinities for the three receptors than that of three carbon derivatives (**6j** vs. **6l**; **9a** vs. **9b**; **9g** vs. **9h**), for example, elongation of the linker of **9c** from three carbons to four carbons resulted in improved activities for D_2_ and 5-HT_1A_ but had a negative effect on the 5-HT_2A_ receptor (**9c** vs. **9d**). This indicated that the appropriate length of the linker was four carbons.

Overall, compounds **6j**, **9d**, **9g** and **9h** demonstrated high affinity for the D_2_, 5-HT_1A_ and 5-HT_2A_ receptors (D_2_, *K*_i_ < 20 nM; 5-HT_1A_, *K*_i_ <10 nM; 5-HT_2A_, *K*_i_ < 10 nM) and the potency ratio between D_2_ and 5-HT_1A_, 5-HT_2A_ was less than 5, which means the compounds have balanced receptor activity profiles. Therefore, compounds **6j**, **9d**, **9g** and **9h** were selected for further evaluation.

A vast number of clinical studies has demonstrated that atypical antipsychotics can induce a series of adverse effects in the treatment of schizophrenia, such as weight gain, hyperglycaemia and QTc prolongation [18,19,20]. These adverse side effects have been proved to be associated with several off-target receptors, mainly focused on histamine H_1_, 5-HT_2C_, and adrenergic α_1_ [21,22]. For instance, the synergistic effects of H_1_ and 5-HT_2C_ antagonism may cause weight gain and hyperglycaemia while antagonism on the adrenergic α_1_ receptor may trigger orthostatic hypotension [23]. Therefore, the selected compounds were further evaluated for these receptors in the present study. As shown in Table 3, compounds **6j**, **9d**, **9g** and **9h** exhibited moderate activities for the α_1_ and H_1_ receptors and lower activities for 5-HT_2c_ receptors compared with risperidone (α_1_, *K*_i_ = 2.8 ± 0.4 nM; H_1_, *K*_i_ = 26.1 ± 2.8 nM; 5-HT_2_c, *K*_i_ = 19.7 ± 2.7 nM), suggesting that **6j**, **9d**, **9g** and **9h** had low liability to elicit treatment-associated adverse effects. In addition, the selected compounds were subjected to additional studies of binding to the D_3_ receptor, as the effect of atypical antipsychotic drugs on the D_3_ receptor is related to cognitive and motivational behaviors [24], and several recent pharmacological studies strongly support the idea that antagonism on the D_3_ receptor might reduce catalepsy [25]. As shown in Table 3, compounds **6j**, **9d**, **9g** and **9h** had high affinities for the D_3_ receptor, which may be beneficial for the treatment of schizophrenia.

Cardiotoxicity is an important drug evaluation indicator, which is often caused by the blockade of the human ether-a-go-go-related gene (hERG) potassium channel [26,27]. To assess the cardiotoxicity of compounds **6j**, **9d**, **9g** and **9h,** their inhibitory actions on hERG were assessed in vitro. As shown in Table 3, all of the tested compounds exhibited lower levels of inhibition on hERG compared to clozapine and risperidone, indicating that they have low risks for QT interval prolongation.

To sum up, all the in vitro evaluations results have shown that compound **6j** exhibited higher affinity for D_2_, D_3_, 5-HT_1A_ and 5-HT_2A_ receptors and weaker affinity for 5-HT_2c_, α_1_ and H_1_ receptors than other candidates, therefore, compound **6j** was then subjected to further investigation.

### 2.3. Acute Toxicity

To assess the safety of compound **6j,** its acute toxicity was assayed in vivo in terms of LD_50_ values. As shown in Table 4, compound **6j** showed a high safety threshold, even exceeds the maximum dose (LD_50_ > 2000 mg/kg). The results indicated that compound **6j** possesses high safety performance and low acute toxicity.

### 2.4. Intrinsic Activity of Compound **6j**

As shown in Table 5, compound **6j** stimulated the D_2L,_ D_3_, 5-HT_1A_, 5-HT_2A_ receptors in the agonist assay and showed weak agonist activity, the efficacy of the reference compounds less than 10%, respectively. In the antagonist assay, the efficacy of compound **6j** blocked the four receptors more than 90%. Thus, **6j** functioned as an antagonist at the D_2L_ (IC_50_ = 8.9 nM), D_3_ (IC_50_ = 31.5 nM), 5-HT_1A_ (IC_50_ = 201.4 nM), 5-HT_2A_ (IC_50_ = 195.5 nM) receptors.

### 2.5. Behavioral Studies

As schizophrenic patients show many psychiatric and somatic symptoms in clinical manifestations, more intuitive and visual models to evaluate the antipsychotic-like activities of the new compounds are required. Nowadays, there are several models commonly used to assess the antipsychotic-like efficacy of potential compounds, such as apomorphine (APO)-induced hyperlocomotion [28,29] and MK-801-induced hyperactivity models [30]. In the APO-induced hyperlocomotion model, the D_2_ receptor agonist apomorphine is commonly used to induce behavioral agitation in mice to simulate the positive symptoms of schizophrenia, but this can be reversed by compounds with antipsychotic-like efficacies. In the MK-801-induced hyperactivity model, the non-competitive NMDA receptor antagonist MK-801 can induce the schizophrenia-like symptoms in healthy animals, but compounds with antipsychotic-like efficacies can significantly reverse these symptoms. In addition, the catalepsy test model is used to predict the propensity of antipsychotics to induce EPS in humans [31]. This model has become a common and important animal model for assessing the potential therapeutic effect in schizophrenia [31]. In this study, **6j** was subjected to these models to verify its potential antipsychotic activity.

#### 2.5.1. Apomorphine-Induced Hyperlocomotion

As shown in Figure 2, compound **6j** induced a dose-dependent response in APO-induced hyperlocomotion model, with an ED_50_ value of 0.19 mg/kg (Table 4). Mice in the control groups received clozapine and risperidone inhibit the APO-induced climbing with ED_50_ values of 17.92 and 0.046 mg/kg, respectively. The comparison of results indicates that compound **6j** has certain efficacy against the positive symptoms of psychosis.

#### 2.5.2. MK-801-Induced Hyperactivity

In the MK-801-induced hyperactivity model, compound **6j** significantly and dose dependently attenuated the increased locomotor activity (Figure 3) with an ED_50_ value of 0.16 mg/kg (Table 4). In the control group, mice treated with clozapine and risperidone with ED_50_ values of 2.28 and 0.011 mg/kg, respectively. The comparison of the results shows that compound **6j** was more potent than clozapine.

#### 2.5.3. Catalepsy

In the catalepsy test model, compound **6j** displayed a high threshold for catalepsy with an ED_50_ value of 56.84 mg/kg, but the ED_50_ values of clozapine and risperidone were 50.0 mg/kg and 0.92 mg/kg, the comparison of results indicates that **6j** has a lower incidence of EPS (Table 4).

In addition, compound **6j** displayed a wide therapeutic index (TI) range (299.16–352.5) based on its efficacy (APO and MK-801 models) and its side effects (catalepsy), while the TI of clozapine range from 5.58 to 21.93, and 20.0 to 83.63 of risperidone (Table 4). The result demonstrated that **6j** has a higher safety margin than clozapine and risperidone.

## 3. Materials and Methods

### 3.1. General Information

All solvents were from commercial sources and used without further purification. The purity of all the reagents and the test compounds was more than 95%. Melting points were determined in open capillary tubes and are uncorrected. ^1^H-NMR spectra were recorded on an Avance III 400 spectrometer (Bruker, Karlsruhe, Germany) at 400 MHz (^1^H) using CDCl_3_ and DMSO-d_6_ as solvents. Chemical shifts recorded as δ values (ppm), using tetramethylsilane (TMS) as the internal reference; Coupling constants (*J*) are given in Hz. Proton multiplicities are labeled as s (singlet), d (doublet), t (triplet), q (quartet), m (multiplet), and br (broad signal). Thin layer chromatography (TLC) was performed using silica gel GF254. Flash column chromatography was carried out using silica gel.

### 3.2. Instrumentation

High performance liquid chromatography (HPLC) methods: LC-20AD spectrometer (Shimadzu, Kyoto, Japan); column, Shimadzu VP-ODS (4.6 mm × 250 mm, 5 μm) C_18_-253; mobile phase, A: acetonitrile 0.01 mol/L NH_4_H_2_PO_4_ (0.2% Et_3_N, pH = 3.0) (10:90); B: acetonitrile 0.01 mol/L NH_4_H_2_PO_4_ (0.1% Et_3_N, pH = 3.0)(70:30); flow rate, 1.0 mL/min; sample size, 10 μL; column temperature, 40 °C; UV detection condition, 210 nm; High resolution mass spectra (HRMS) spectrometer, Agilent 6530 Q-TOF LC/MS and Agilent 1290 Infinity (G4212A) spectrometer (Agilent, Santa Carla, CA, USA). The HPLC conditions as following: column X Bridge^R^ Shield RP18 (4.6 × 150 mm, 3.5 μm, Waters, Milford MA USA), C18-330; mobile phase: A, 5mmol/L NH_4_OAc (pH = 6.0)(10:90); B, MeOH; eluent, 30% A and 70% B (V:V); flow rate, 1.0 mL/min; column temperature, 35 °C; UV detection, 254 nm. Mass spectrometry conditions: Dual AJS ESI, positive; Gas temperature, 350 °C; Fragmentor, 80 KV; Drying gas, 7 mL/min; Nebulizer, 45 psi; Sheath gas temperature, 350 °C; Capillary Voltage, 4000 V; Sheath gas flow, 11 L/min; Nozzle Voltage, 500 V. Low resolution mass spectrometry (LRMS) and high performance liquid chromatography-mass spectrometry (HPLC-MS) were obtained using an Agilent MS/1200 HPLC liquid chromatograph/mass spectrometer (JXZX-FXS-064).Agilent ZORBAX SB-C18 (4.6 × 150 mm, 5 µm); ESI source: scan mode, positive/negative; Rage, 50–1000; Nitrogen gas flow, 11 L/min; Column pressure: −3000–3000 eV. Chromatographic parameter: Mobile phase, 50% acetonitrile, 50% water; UV detection, 210 nm, 245 nm; injection volume, 1.0 µL; flow rate, 0.3 mL/min; column temperature, 30 °C.

### 3.3. Synthesis

#### 3.3.1. General Procedures for the Preparation of Intermediates **4** and **7**

Compound **4**, **7a** and **7b** were synthesized by the following reaction: 2,4-dihydroxyacetophenone and pyrrolidine were added to a suitable amount of acetonitrile and then corresponding ketone was added and the mixture was stirred at 50 °C for 12 h. Then, it was cooled down and poured into an ice-cold 2 M solution of hydrochloric acid and water, stirred for another 30 min, filtered and the obtained solid was recrystallized from EtOH (95%) to give **4** and **7a**–**7b**.

#### 3.3.2. General Procedures for the Preparation of Intermediates 5 and 8

To a suspension of intermediate **4** and **7** in DMF, 1.5 equivalents of K_2_CO_3_ and 1-bromo-3-chloropropane or 1-bromo-4-chlorobutane were added and the suspension was stirred for 24 h at room temperature and then filtered. The filtrate was evaporated in vacuo followed by addition of dichloromethane (DCM), washed with water and dried over anhydrous Na_2_SO_4_, the solvent was evaporated in vacuo and the residue was separated by silica gel column chromatography to afford **5a**, **5b** and **8a**–**8d**.

#### 3.3.3. General Procedures for the Preparation of Target Compounds 6 and 9

A mixture of intermediates **5** or **8**, anhydrous potassium carbonate, CH_3_CN and a catalytic amount of potassium iodide (KI) was stirred at 70 °C for 10 h. The mixture was filtered and the filtrate was concentrated under reduced pressure, the residue was dissolved in DCM, washed with water and dried with anhydrous MgSO_4_, the solvent was removed under reduced pressure and the residue was separated by silica gel column chromatography to afford **6a**-**6l** and **9a**-**9k**.

#### 3.3.4. General Procedures for the Preparation of Intermediates **11, 12, 14,** and **15**

A mixture of resorcinol or a derivative, 3-chloropropionic acid and trifluoromethanesulfonic acid was stirred at 80 °C for 1 h. The reaction mixtures were cooled to room temperature and poured into water, extracted with ethyl acetate (EA), The combined organic phase was washed with water and dried over anhydrous MgSO_4_, the solvent was removed under reduced pressure and the residue was added to a certain amount of sodium hydroxide solution (2N) at 0–5 °C, then stirred for 2 h. The pH was adjusted to above 2 with hydrochloric acid solution and stand for 30 min, the formed intermediates **11a**–**11d** were collected by filtration, washed with water and dried in vacuo.

A mixture of resorcinol or 2-chlororesorcinol, crotonic acid and anhydrous zinc chloride was stirred at 180 °C for 30 min. The reaction mixtures were cooled to room temperature and poured into water, extracted with EA, the combined organic layer was washed with water, brine, and dried over MgSO_4_, the solvents were removed under reduced pressure and the crude products was purified via column chromatography to give pure products **14a**–**14b**.

Intermediates **12** and **15** were prepared as described for compound **5** using **11** or **14**, 1-bromo-4-chlorobutane and anhydrous K_2_CO_3_ in DMF; the crude products were purified via column chromatography.

#### 3.3.5. General Procedures for the Preparation of Target Intermediates 13 and 16

The target compounds **13a**–**13d** and **16a**–**16b** were prepared as described for compound **6** using **12a**–**12e**, **15a**–**15b** and K_2_CO_3_, KI, 6-fluoro-3-(piperidin-4-yl) benzo[d]isoxazole in acetonitrile, the crude products were purified via column chromatography.

#### 3.3.6. General Procedures for the Preparation of Intermediates **1****7, 18** and **19**

1-(2,4-Dihydroxyphenyl)propan-1-one (16.6 g) and sodium acetate anhydrous (8.3 g) were added to acetic anhydride (70 mL), the reaction mixture was stirred at reflux for 14 h and cooled to room temperature, diluted with water and extracted with DCM, the organic layer was separated and washed with water, dried over MgSO_4_, concentrated in vacuo to give intermediate **1****7** as a yellow solid. The intermediate **17** was dissolved in a solution (50 mL) of saturated sodium bicarbonate and MeOH (V:V = 1:1), the suspension was stirred at rt for 3 h, the product intermediate **18** obtained by filtration. Intermediate **18** was dissolved in MeOH, and then Pd/C added and stirred at rt for 10 h under hydrogen atmosphere, filtered and the filtrate concentrated in vacuo to give intermediate **19.**

#### 3.3.7. The Preparation of Intermediate **20** and Target Compound **21**

The intermediate **20** and target compound **21** were prepared as described for intermediate **15** and target compound **16**.

#### 3.3.8. The Characteristics and Spectroscopic Data of the Target Compounds

*2,2-Dimethyl-7-(4-(4-phenylpiperazin-1-yl)butoxy) chroman-4-one* (**6a**).Pale-white solid; m. p. 131–133 °C; yield 80.3%; ^1^H-NMR (CDCl_3_) δ 7.84 (d, *J* = 8.8 Hz, 1H), 7.32 (dd, *J* = 8.6, 7.4 Hz, 2H), 6.98 (dd, *J* = 8.8, 0.9 Hz, 2H), 6.91 (t, *J* = 7.3 Hz, 1H), 6.58 (dd, *J* = 8.8, 2.4 Hz, 1H), 6.42 (d, *J* = 2.3 Hz, 1H), 4.07 (t, *J* = 6.3 Hz, 2H), 3.31–3.21 (m, 4H), 2.71 (s, 2H), 2.70–2.65 (m, 4H), 2.56–2.43 (m, 2H), 1.89 (dd, *J* = 14.4, 6.3 Hz, 2H), 1.83–1.69 (m, 2H), 1.50 (d, *J* = 2.0 Hz, 6H). HRMS (ESI) calculated for C_25_H_33_N_2_O_3_ [M + H]^+^, 409.2486; found, 409.2479.

*7-(4-(4-(2-Fluorophenyl) piperazin-1-yl) butoxy)-2,2-dimethylchroman-4-one* (**6b**). Pale-white solid; m. p. 135–137 °C; yield 78.8%; ^1^H-NMR (CDCl_3_) δ 7.84 (d, *J* = 8.8 Hz, 1H), 7.19–6.87 (m, 4H), 6.57 (dd, *J* = 8.8, 2.3 Hz, 1H), 6.41 (d, *J* = 2.3 Hz, 1H), 4.06 (t, *J* = 6.3 Hz, 2H), 3.27–3.06 (m, 4H), 2.71–2.97 (m, 6H), 2.58–2.40 (m, 2H), 1.93–1.71 (m, 4H), 1.49 (s, 6H). HRMS (ESI) calculated for C_25_H_32_FN_2_O_3_ [M + H]^+^, 427.2391; found, 427.2390.

*2,2-Dimethyl-7-(4-(4-(3-(trifluoromethyl) phenyl) piperazin-1-yl) butoxy) chroman-4-one* (**6c**). Pale-white solid; m. p. 139–140 °C; yield 76.2%; ^1^H-NMR (CDCl_3_) δ 7.84 (d, *J* = 8.8 Hz, 1H), 7.40 (t, *J* = 8.0 Hz, 1H), 7.31 (s, 1H), 7.20 - 7.02 (m, 2H), 6.58 (dd, *J* = 8.8, 2.4 Hz, 1H), 6.41 (dd, *J* = 5.0, 2.7 Hz, 1H), 4.07 (dd, *J* = 8.0, 4.4 Hz, 2H), 3.33 (s, 4H), 2.68 (d, *J* = 24.9 Hz, 6H), 2.56 (s, 2H), 1.96–1.85 (m, 2H), 1.83–1.71 (m, 2H), 1.50 (d, *J* = 2.7 Hz, 6H). HRMS (ESI) calculated for C_26_H_32_F_3_N_2_O_3_ [M + H]^+^, 477.2360; found, 477.2358.

*7-(4-(4-(2-Methoxyphenyl) piperazin-1-yl) butoxy)-2,2-dimethylchroman-4-one* (**6d**).Off-white solid; m. p. 128–130 °C; yield 75.5%; ^1^H-NMR (CDCl_3_) δ 7.80 (t, *J* = 4.0 Hz, 1H), 7.02–6.87 (m, 4H), 6.57 (m, 1H), 6.41 (d, *J* = 4.0 Hz, 1H), 4.09 (d, *J* = 4.0 Hz, 2H), 3.87 (s, 2H), 2.70-2.60 (m, 4H), 2.05 (d, *J* = 8.0 Hz, 2H), 2.68 (d, *J* = 22.9 Hz, 2H), 2.67–2.58 (m, 4H), 2.57–2.42 (m, 2H), 1.89 (dd, *J* = 14.4, 6.3 Hz, 2H), 1.45 (s, 6H). HRMS (ESI) calculated for C_26_H_34_N_2_O_4_ [M + H]^+^, 438.2519; found, 438.2511.

*2,2-Dimethyl-7-(4-(4-(pyridin-2-yl) piperazin-1-yl) butoxy) chroman-4-one* (**6e**).Pale-white solid; m. p. 127–129 °C; yield 73.2%; ^1^H-NMR (CDCl_3_) δ 8.33 - 8.17 (m, 1H), 7.83 (t, *J* = 8.1 Hz, 1H), 7.53 (ddd, *J* = 8.9, 7.1, 2.0 Hz, 1H), 6.79–6.62 (m, 2H), 6.58 (dd, *J* = 8.8, 2.4 Hz, 1H), 6.42 (d, *J* = 2.3 Hz, 1H), 4.07 (t, *J* = 6.3 Hz, 2H), 3.67–3.53 (m, 4H), 2.68 (d, *J* = 22.9 Hz, 2H), 2.67–2.58 (m, 4H), 2.57–2.42 (m, 2H), 1.89 (dd, *J* = 14.4, 6.3 Hz, 2H), 1.78 (dd, *J* = 14.9, 8.1 Hz, 2H), 1.49 (s, 6H). HRMS (ESI) calculated for C_24_H_32_N_3_O_3_ [M + H]^+^, 410.2438; found, 410.2438.

*2,2-Dimethyl-7-(4-(4-(pyrimidin-2-yl) piperazin-1-yl) butoxy) chroman-4-one* (**6f**).Pale-white solid; m. p. 128–131 °C; yield 69.7%; ^1^H-NMR (CDCl_3_) δ 8.35 (d, *J* = 4.7 Hz, 2H), 7.83 (d, *J* = 8.8 Hz, 1H), 6.64–6.52 (m, 2H), 6.42 (d, *J* = 2.3 Hz, 1H), 4.06 (t, *J* = 6.3 Hz, 2H), 3.90 (s, 4H), 2.71 (s, 2H), 2.56 (d, *J* = 20.1 Hz, 4H), 2.51 (d, *J* = 7.4 Hz, 2H), 1.94 - 1.82 (m, 2H), 1.82 - 1.70 (m, 2H), 1.49 (s, 6H). HRMS (ESI) calculated for C_23_H_31_N_4_O_3_ [M + H] ^+^, 411.2391; found, 411.2390.

*7-(4-(4-(Benzo[d]* [1,3] *dioxol-5-ylmethyl) piperazin-1-yl) butoxy)-2,2-dimethylchroman-4-one*(**6g**).White solid; m. p. 112–114 °C; yield 69.4%; ^1^H-NMR (CDCl_3_) δ 7.82 (d, *J* = 8.8 Hz, 1H), 6.89 (s, 1H), 6.78 (d, *J* = 0.8 Hz, 2H), 6.55 (dd, *J* = 8.8, 2.3 Hz, 1H), 6.39 (d, *J* = 2.3 Hz, 1H), 5.97 (s, 2H), 4.16 (q, *J* = 7.1 Hz, 2H), 4.03 (t, *J* = 6.4 Hz, 2H), 3.45 (s, 2H), 2.70 (s, 2H), 2.46 (m, 6H), 1.94–1.60 (m, 4H), 1.49 (s, 6H), 1.30 (t, *J* = 7.1 Hz, 2H). HRMS (ESI) calculated for C_27_H_35_N_2_O_5_ [M + H]^+^, 467.2540; found, 467.2529.

*7-(4-(4-(Benzo[b]thiophen-4-yl) piperazin-1-yl) butoxy)-2,2-dimethylchroman-4-one*(**6h**).Pale white solid; m. p. 117119 °C; yield 67.7%; ^1^H-NMR (CDCl_3_) δ 7.85 (d, *J* = 8.8 Hz, 1H), 7.60 (d, *J* = 8.1 Hz, 1H), 7.45 (q, *J* = 5.6 Hz, 2H), 7.32 (dd, *J* = 9.6, 6.1 Hz, 1H), 6.95 (d, *J* = 7.6 Hz, 1H), 6.59 (dd, *J* = 8.8, 2.3 Hz, 1H), 6.43 (d, *J* = 2.3 Hz, 1H), 4.27 - 3.97 (m, 2H), 3.26 (s, 4H), 2.75 (d, *J* = 30.9 Hz, 4H), 2.64–2.41 (m, 2H), 2.02–1.70 (m, 4H), 1.50 (s, 6H), 1.31 (t, *J* = 7.1 Hz, 2H). HRMS (ESI) calculated for C_27_H_33_N_2_O_3_S [M + H]^+^, 465.2206; found, 465.2201.

*7-(4-(4-(Benzo[d]isothiazol-3-yl) piperazin-1-yl) butoxy)-2,2-dimethylchroman-4-one* (**6i**). Pale white solid; m. p. 119–120 °C; yield 65.8%; ^1^H-NMR (CDCl_3_) δ 7.96 (d, *J* = 8.2 Hz, 1H), 7.85 (t, *J* = 8.1 Hz, 2H), 7.52 (ddd, *J* = 8.1, 7.0, 1.0 Hz, 1H), 7.41 (ddd, *J* = 8.0, 7.0, 0.9 Hz, 1H), 6.58 (dd, *J* = 8.8, 2.4 Hz, 1H), 6.43 (d, *J* = 2.3 Hz, 1H), 4.08 (t, *J* = 6.3 Hz, 2H), 3.71–3.50 (m, 4H), 2.86–2.66 (m, 4H), 2.64–2.44 (m, 2H), 1.98–1.68 (m, 4H), 1.50 (s, 6H), 1.31 (t, *J* = 7.1 Hz, 2H). HRMS (ESI) calculated for C_26_H_32_N_3_O_3_S [M + H]^+^, 466.2195; found, 466.2189.

*7-(4-(4-(6-Fluorobenzo[d]isoxazol-3-yl) piperidin-1-yl) butoxy)-2,2-dimethylchroman- 4-one* (**6j**). Pale white solid; m. p. 116–117 °C; yield 70.2%; ^1^H-NMR (CDCl_3_) δ 7.80 (d, *J* = 8.8 Hz, 1H), 7.70 (dd, *J* = 8.7, 5.0 Hz, 1H), 7.25 (dd, *J* = 8.5, 1.9 Hz, 1H), 7.06 (td, *J* = 8.9, 2.1 Hz, 1H), 6.57 (dd, *J* = 8.8, 2.4 Hz, 1H), 6.40 (d, *J* = 2.4 Hz, 1H), 4.23–4.13 (m, 2H), 3.86 (dd, *J* = 5.4, 4.2 Hz, 2H), 3.74 (d, *J* = 5.8 Hz, 2H), 3.15 – 3.01 (m, 3H), 2.69 (dd, *J* = 11.2, 5.4 Hz, 4H), 2.26 (dd, *J* = 11.3, 3.0 Hz, 2H), 2.15 – 2.02 (m, 4H), 1.45 (s, 6H). HRMS (ESI) calculated for C_27_H_32_FN_2_O_4_ [M + H]^+^, 467.2341; found, 467.2335.

*7-(3-(4-(2-Methoxyphenyl)piperazin-1-yl)propoxy)-2,2-dimethylchroman-4-one* (**6k**). Pale-white solid; m. p. 103–105 °C; yield 71.5%; ^1^H-NMR (CDCl_3_) δ 7.80 (d, *J* = 8.0 Hz, 1H), 7.02–6.87(m, 4 H), 6.54–6.57 (m, 1H), 6.40 (d, *J* =4 Hz, 1H), 3.88 (s, 3H), 3.13 (s, br, 4H), 2.70–2.67 (m, 8H), 2.60 (t, *J* = 8.0 Hz, 2H), 2.05-2.03 (m, 2H), 1.46 (s, 6H). HRMS (ESI) calculated for C_25_H_33_FN_2_O_4_ [M + H]^+^, 425.2435; found, 425.2433.

*7-(3-(4-(6-Fluorobenzo[d]isoxazol-3-yl)piperidin-1-yl)propoxy)-2,2-dimethylchroman-4-one* (**6l**). Pale-white solid; m. p. 116–118 °C; yield 73.4%; ^1^H-NMR (CDCl_3_) δ 7.80 (d, *J* = 8.0 Hz, 1H), 7.74–7.70 (m, 1H), 7.27–7.25 (m, 1H), 7.09-7.05 (m, 1H), 6.57-6.54 (m, 1H), 6.40 (d, *J* = 4.0 Hz, 1H), 4.10 (t, *J* = 8.0 Hz, 2H), 3.13-3.09 (m, 3H), 2.68 (s, 2H), 2.59 (t, *J* = 8.0 Hz, 2H), 2.21-2.00 (m, 8H), 1.46 (s, 6H).HRMS (ESI) calculated for C_26_H_30_FN_2_O_4_ [M + H]^+^, 453.2184; found, 453.2182.

*7-(3-(4-(2-Methoxyphenyl) piperazin-1-yl) propoxy) spiro[chromane-2,1′-cyclopentan]-4-one* (**9a**). Off-white solid; m. p. 128–130 °C; yield 68.5%; ^1^H-NMR (CDCl_3_) δ 7.81 (d, *J* = 8.0 Hz, 1H), 7.04–6.88(m, 4H), 6.58–6.55 (m, 1H), 6.40 (d, *J* = 4.0 Hz, 1H), 4.09 (t, *J* = 8.0 Hz, 2H), 3.89 (s, 3H), 3.13 (s, br, 4H), 2.79 (s, 2H), 2.71 (s, br, 4H), 2.61 (t, *J* = 8.0 Hz, 2H), 2.03–2.12 (m, 4H), 1.87-1.91 (m, 2H), 1.74-1.65 (m, 4H). HRMS (ESI) calculated for C_27_H_35_FN_2_O_4_ [M + H]^+^, 451.2519; found, 451.2511.

*7-(4-(4-(2-Methoxyphenyl) piperazin-1-yl) butoxy) spiro[chromane-2,1′-cyclopentan]- 4-one* (**9b**). Off-white solid; m. p. 96–98 °C; yield 76.4%; ^1^H-NMR (CDCl_3_) δ 7.80 (d, *J* = 8.0 Hz, 1H), 7.03–6.86 (m, 4H), 6.56–6.53 (m, 1H), 6.37 (d, *J* = 4.0 Hz, 1H), 4.03 (t, *J* = 8.0 Hz, 2H), 3.88 (s, 3H), 3.12 (s, br, 4H), 2.78 (s, 2H), 2.68 (s, br, 4H), 2.49 (t, *J* = 8.0 Hz, 2H), 2.10–2.05 (m, 2H), 1.87–1.84 (m, 4H), 1.75–1.69 (m, 6H). HRMS (ESI) calculated for C_27_H_35_FN_2_O_4_ [M + H]^+^, 451.2519; found, 451.2511. HRMS (ESI) calculated for C_28_H_37_N_2_O_4_ [M+H]^+^, 465.2748; found, 465.2745.

*7-(3-(4-(6-Fluorobenzo[d]isoxazol-3-yl)piperidin-1-yl)propoxy)spiro[chromane-2,1′-cyclopentan]-4-one* (**9c**). White solid; m. p. 91–93 °C; yield 70.2%; ^1^H-NMR (CDCl_3_) δ 7.80 (d, *J* = 8.0 Hz, 1H), 7.72-7.69(m, 1H), 7.26–7.24 (m, 1H), 7.09-7.04 (m, 1H), 6.57–6.54(m, 1H), 6.41 (d, *J* = 4.0 Hz, 1H), 4.09 (t, *J* = 8.0 Hz, 2H), 3.11-3.08 (m, 3H), 2.78 (s, 2H), 2.58 (t, *J* = 8.0 Hz, 2H), 2.23-1.86 (m, 12H), 1.74-1.64 (m, 4H). HRMS (ESI) calculated for C_28_H_32_FN_2_O_4_ [M + H]^+^, 479.2341; found, 479.2338.

*7-(4-(4-(6-Fluorobenzo[d]isoxazol-3-yl)piperidin-1-yl)butoxy)spiro[chromane-2,1′-cyclopentan]-4-one* (**9d**). Pale-white solid; m. p. 103–105 °C; yield 68.6%; ^1^H-NMR (CDCl_3_) δ 7.80 (d, *J* = 8.0 Hz, 1H), 7.69–7.73(m, 1H), 7.26–7.24 (m, 1H), 7.09–7.04 (m, 1H), 6.56–6.54 (m, 1H), 6.38 (d, *J* = 4.0 Hz, 1H), 4.04 (t, *J* = 8.0 Hz, 2H), 3.12-3.08 (m, 3H), 2.78 (s, 2H), 2.48 (t, *J* = 8.0 Hz, 2H), 2.19-2.07 (m, 8H), 1.90-1.63 (s, 10H). HRMS (ESI) calculated for C_29_H_34_FN_2_O_4_ [M + H]^+^, 493.2497; found, 493.2496.

*7-(3-(4-(2-Methoxyphenyl)piperazin-1-yl)propoxy)spiro[chromane-2,1′-cyclohexan]-4-one* (**9e**). Pale-white solid; m. p. 105–107 °C; yield 71.4%; ^1^H-NMR (CDCl_3_) δ 7.79 (d, *J* = 8.0 Hz, 1H), 6.85–7.00 (m, 4H), 6.56–6.53 (m, 1H), 6.44 (d, *J* = 4.0 Hz, 1H), 4.09 (t, *J* = 8.0 Hz, 2H), 3.87 (s, 3H), 3.12 (m, 4H), 2.70 (s, br, 4H), 2.64 (s, 2H), 2.60 (t, *J* = 8.0 Hz, 2H), 2.07-1.97 (m, 4H), 1.72-1.66 (m, 2H), 1.52-1.45 (m, 4H), 1.34–1.30 (s, 2H). HRMS (ESI) calculated for C_28_H_37_N_2_O_4_ [M + H]^+^, 465.2748; found, 465.2740.

*7-(3-(4-(2-(Trifluoromethyl)phenyl)piperazin-1-yl)propoxy)spiro[chromane-2,1′-cyclohexan]-4-one* (**9f**). Off-white solid; m. p. 138–140 °C; yield 74.7%; ^1^H-NMR (CDCl_3_) δ 7.80–7.73 (m, 1H), 7.42–7.34 (m, 1H), 7.19–7.06 (m, 3H), 6.56–6.52 (m, 1H), 6.41 (d, *J* = 4.0 Hz, 1H), 4.05 (t, *J* = 8.0 Hz, 2H), 3.31-3.28 (m, 2H), 2.70–2.52 (m, 6H), 2.30–2.05 (m, 4H), 1.99–1.48 (m, 10H), 1.36–1.32 (m, 2H). HRMS (ESI) calculated for C_28_H_34_F_3_N_2_O_3_ [M + H]^+^, 503.2516; found, 503.2511.

*7-(3-(4-(6-Fluorobenzo[d]isoxazol-3-yl)piperidin-1-yl)propoxy)spiro[chromane-2,1′-cyclohexan]-4-one* (**9g**). Pale-white solid; m. p. 96–98 °C; yield 67.1%; ^1^H-NMR (CDCl_3_) δ 7.79(d, *J* = 8.0 Hz, 1H), 7.74–7.70 (m, 1H), 7.26–7.24 (m, 1H), 7.10–7.05 (m, 1H), 6.58-6.55 (m, 1H), 6.45 (d, *J* = 4.0 Hz, 1H), 4.11 (t, *J* = 8.0 Hz, 2H), 3.12–3.10 (m, 3H), 2.66 (s, 2H), 2.61 (t, *J* = 8.0 Hz, 2H), 2.2–1.98 (m, 12H), 1.74–1.48 (m, 4H), 1.36-1.33 (m, 2H). HRMS (ESI) calculated for C_29_H_34_FN_2_O_4_ [M + H]^+^, 493.2497; found, 493.2495.

*7-(4-(4-(6-Fluorobenzo[d]isoxazol-3-yl)piperidin-1-yl)butoxy)spiro[chromane-2,1′-cyclohexan]-4-one* (**9h**). White solid; m. p. 126–128 °C; yield 72.9%; ^1^H-NMR (CDCl_3_) δ 7.78(d, *J* = 8.0 Hz, 1H), 7.70–7.73 (m, 1H), 7.26–7.24 (m, 1H), 7.09–7.04 (m, 1H), 6.55–6.52 (m, 1H), 6.42 (d, *J* = 4.0 Hz, 1H), 4.05 (t, *J* = 8.0 Hz, 2H), 3.13-3.06 (m, 3H), 2.65 (s, 2H), 2.49(t, *J* = 8.0 Hz, 2H), 2.18-1.98 (m, 8H), 1.86-1.84 (m, 2H), 1.75-1.69 (m, 4H), 1.53-1.50 (m, 4H), 1.37-1.31 (m, 2H). HRMS (ESI) calculated for C_30_H_36_FN_2_O_4_ [M + H]^+^, 507.2654; found, 507.6259.

*7-(4-(4-(2,3-Dimethylphenyl)piperazin-1-yl)butoxy)spiro[chromane-2,1′-cyclohexan]-4-one* (**9i**). White solid; m. p. 138–140 °C; yield 74.3%; ^1^H-NMR (CDCl_3_) δ 7.81–7.76 (m, 1H), 7.10–6.92(m, 3H), 6.56–6.52 (m, 1H), 6.42 (d, *J* = 4.0 Hz, 1H), 4.06(t, *J* = 8.0 Hz, 2H), 3.06 (s, br, 4H), 2.66 (s, 2H), 2.60 (t, *J* = 8.0 Hz, 2H), 2.29 (s, 3H), 2.23 (s, 3H), 2.07–1.48 (m, 16H), 1.37–1.31 (m, 2H).HRMS (ESI) calculated for C_30_H_41_N_2_O_3_ [M + H]^+^,477.3112; found, 477.3110.

*7-(4-(4-(2,3-Dichlorophenyl)piperazin-1-yl)butoxy)spiro[chromane-2,1′-cyclohexan]-4-one*(**9j**). White solid; m. p. 119–121 °C; yield 70.7%; ^1^H-NMR (CDCl_3_) δ 7.81–7.76(m, 1H), 7.10–6.92 (m, 3H), 6.56–6.52 (m, 1H), 6.42 (d, *J* = 4.0 Hz, 1H), 4.06 (t, *J* = 8.0 Hz, 2H), 3.00 (s, br, 4H), 2.66 (s, 2H), 2.54(t, *J* = 8.0 Hz, 3H), 2.29 (s, 3H), 2.01-1.48 (m, 12H), 1.37–1.33 (m, 2H). HRMS (ESI) calculated for C_28_H_35_Cl_2_N_2_O_3_ [M + H]^+^, 517.2019; found, 517.2013.

*7-(4-(4-(Benzo[d]isothiazol-3-yl)piperidin-1-yl)butoxy)spiro[chromane-2,1′-cyclohexan]-4-one* (**9k**). Pale-white solid; m. p. 129–130 °C; yield 62.8%; ^1^H-NMR (CDCl_3_) δ 7.79 (d, *J* = 8.0 Hz, 2H), 7.19–7.16 (m, 2H), 7.49–7.45 (m, 1H), 6.99–6.96 (m, 1H), 6.56–6.53 (m, 1H), 6.42 (d, *J* = 4.0 Hz, 1H), 4.05 (t, *J* = 8.0 Hz, 2H), 3.11(s, br, 4H), 2.70 (s, br, 4H), 2.66(s, 2H), 2.52 (t, *J* = 8.0 Hz, 2H), 2.00-1.48 (m, 12H), 1.34-1.31 (m, 2H). HRMS (ESI) calculated for C_30_H_37_N_2_O_3_S [M + H]^+^, 505.2519; found, 505.2514.

*7-(4-(4-(6-Fluorobenzo[d]isoxazol-3-yl)piperidin-1-yl)butoxy)chroman-4-one* (**13a**). Colorless oily liquid; yield 58.1%; ^1^H-NMR (CDCl_3_) δ 7.80(d, *J* = 8.0 Hz, 1H), 7.75–7.71 (m, 1H), 7.26–7.24 (m, 1H), 7.11–7.06 (m, 1H), 6.56–6.53 (m, 1H), 6.38 (d, *J* = 4.0 Hz, 1H), 4.42 (t, *J* = 8.0 Hz, 2H), 4.06 (t, *J* = 8.0 Hz, 2H),3.06–2.95 (m, 4H), 2.81–2.78 (m, 1H), 2.52–2.43 (m, 4H), 1.92–1.69 (m, 6H), 1.41-1.35 (m, 2H). HRMS (ESI) calculated for C_25_H_28_FN_2_O_4_ [M + H]^+^, 439.2018; found, 439.2017.

*7-(4-(4-(6-Fluorobenzo[d]isoxazol-3-yl)piperidin-1-yl)butoxy)-5-methylchroman-4-one* (**13b**). Pale-white solid; m. p. 85–87 °C; yield 63.6%; ^1^H-NMR (CDCl_3_) δ 7.75–7.71 (m, 1H), 7.25–7.23 (m, 1H), 7.06–7.11 (m, 1H), 6.54–6.57 (m, 1H), 6.39 (d, *J* = 4.0 Hz, 1H), 4.45 (t, *J* = 8.0 Hz, 2H), 4.07 (t, *J* = 8.0 Hz, 2H), 3.07–2.96 (m, 4H), 2.80–2.77 (m, 1H), 2.52–2.43 (m, 4H), 2.39 (s, 3H), 1.91–1.67 (m, 6H), 1.40–1.33 (m, 2H). HRMS (ESI) calculated for C_26_H_30_FN_2_O_4_ [M + H]^+^, 453.2184; found, 453.2182.

*7-(4-(4-(6-Fluorobenzo[d]isoxazol-3-yl)piperidin-1-yl)butoxy)-6-methylchroman*-*4-one* (**13c**). Pale-white solid; m. p. 91–93 °C; yield 60.3%; ^1^H-NMR (CDCl_3_) δ 7.80 (d, *J* = 8.0 Hz, 1H), 7.74–7.70 (m, 1H), 7.26-7.24 (m, 1H), 7.12–7.07 (m, 1H), 6.38 (d, *J* = 4.0 Hz, 1H), 4.46 (t, *J* = 8.0 Hz, 2H), 4.06 (t, *J* = 8.0 Hz, 2H), 3.08–2.99 (m, 4H), 2.80–2.77 (m, 1H), 2.52–2.43 (m, 4H), 2.38 (s, 3H), 1.92–1.68 (m, 6H), 1.41–1.35 (m, 2H). HRMS (ESI) calculated for C_26_H_30_FN_2_O_4_ [M + H]^+^, 453.2184; found, 453.2182.

*7-(4-(4-(6-Fluorobenzo[d]isoxazol-3-yl)piperidin-1-yl)butoxy)-8-methylchroman-4-one* (**13d**). Pale-white solid; m.p. 88–90 °C; yield 67.0%; ^1^H-NMR (CDCl_3_) δ 7.80 (d, *J* = 8.0 Hz, 1H), 7.74–7.70 (m, 1H), 7.26–7.24 (m, 1H), 7.12–7.07 (m, 1H), 6.57–6.54 (m, 1H), 4.45 (t, *J* = 8.0 Hz, 2H), 4.06 (t, *J* = 8.0 Hz, 2H), 3.08–3.01 (m, 4H), 2.82–2.79 (m, 1H), 2.55–2.44 (m, 4H), 2.40 (s, 3H), 1.91–1.69 (m, 6H), 1.41–1.36 (m, 2H). HRMS (ESI) calculated for C_26_H_30_FN_2_O_4_ [M + H]^+^, 453.2184; found, 453.2179.

*7-(4-(4-(6-Fluorobenzo[d]isoxazol-3-yl)piperidin-1-yl)butoxy)-2-methylchroman-4–one* (**16a**). Pale-white solid; m. p. 95–97 °C; yield 63.4%; ^1^H-NMR (CDCl_3_) δ 7.80(d, *J* = 8.0 Hz, 1H), 7.74–7.70 (m, 1H), 7.27–7.25 (m, 1H), 7.10–7.05 (m, 1H), 6.57–6.54 (m,1H), 6.39 (d, *J* = 4.0 Hz, 1H), 4.41–4.38 (m, 1H), 4.05 (t, *J* = 8.0 Hz, 2H), 3.06–3.01 (m, 3H), 2.81–2.78 (m, 2H), 2.51–2.41 (m, 4H), 1.93–1.67 (m, 6H), 1.50–1.47 (m, 5H). HRMS (ESI) calculated for C_26_H_30_FN_2_O_4_ [M + H]^+^, 453.2184; found, 453.2180.

*8-Chloro-7-(4-(4-(6-fluorobenzo[d]isoxazol-3-yl)piperidin-1-yl)butoxy)-2-methylchroman-4-one* (**16b**). Pale-white solid; m. p. 99–101 °C; yield 64.7%; ^1^H-NMR (CDCl_3_) δ 7.81(d, *J* = 8.0 Hz, 1H), 7.74–7.71 (m, 1H), 7.26–7.23 (m, 1H), 7.10–7.05 (m, 1H), 6.56–6.54 (m,1H), 4.40–4.38 (m, 1H), 4.05 (t, *J* = 8.0 Hz, 2H), 3.05–3.01 (m, 3H), 2.81–2.78 (m, 2H), 2.51–2.41 (m, 4H), 1.92–1.67 (m, 6H), 1.50–1.46 (m, 5H). HRMS (ESI) calculated for C_26_H_29_ClFN_2_O_4_ [M + H]^+^, 487.1794; found, 487.1785.

*7-(4-(4-(6-fluorobenzo[d]isoxazol-3-yl)piperidin-1-yl)butoxy)-2,3-dimethylchroman-4-one* (**21**). Pale-white solid; m. p. 82–84 °C; yield 86.1%; ^1^H-NMR (CDCl_3_) δ 7.80 (d, *J* = 8.0 Hz, 1H), 7.75–7.71 (m, 1H), 7.26–7.24 (m, 1H), 7.13–7.08 (m, 1H), 6.58–6.54 (m, 1H), 6.38 (d, *J* = 4.0 Hz, 1H), 4.45 (m, 1H), 4.05 (t, *J* = 8.0 Hz, 2H), 3.69 (m, 1H), 3.03-3.01(m, 2H), 2.82-2.78 (m, 1H), 2.55–2.44 (m, 4H), 1.92–1.69 (m, 6H), 1.41–1.28 (m, 8H). HRMS (ESI) calculated for C_27_H_32_FN_2_O_4_ [M + H]^+^, 467.2341; found, 467.2335.

### 3.4. Biological Studies

#### Ethics Statement

Chinese Kun Ming (KM) Mice (20 ± 2.0 g) and Sprague-Dawley (SD) rats (250 ± 5.0 g) were used as experimental animals in this study. Animals were housed under standardized light and temperature conditions and received standard rat chow and tap water at libitum. Animals were randomly assigned to different experimental groups and each group was kept in a separate cage. All the research involving animals in this study follows the guidelines of the bylaws on experiments on animals, and has been approved by the Ethics and Experimental Animal Committee of Jiangsu Ocean University (Project identification code:2020002, date of approval: 8 January 2020). For the procedural details of the biological studies see the Appendix A.

## 4. Conclusions

In summary, a series of new flavone derivatives were synthesized based on heterocyclic and aromatic piperazines (piperidine), and their antipsychotic activities were evaluated. Of these compounds, **6j** was favorable for binding to multiple acceptors with the KI value of D_2_ (12.1 nM), D_3_ (25.4 nM), 5-HT_1A_ (9.7 nM) and 5-HT_2A_ (3.2 nM), respectively. On the other hand, **6j** showed low potency for 5-HT_2C_, α_1_, H_1_ receptors and human ether-a-go-go-related gene (hERG) channel (IC_50_ 3769.8 nM), which are closely related to the adverse effects of existing antipsychotics. In addition, compound **6j** exhibited significant inhibition of schizophrenia-like symptoms (APO and MK-801-induced motor behavior) without causing catalepsy. Furthermore, compound **6j** with its high value of ED_50_ (58.4 mg/kg) in catalepsy and LD_50_ (>2000 mg/kg) acts at a low dose in apomorphine-induced climbing test (ED_50_ 0.19 mg/kg), which results in a wider therapeutic index range compared with the typical antipsychotics clozapine and risperidone. Thus, compound **6j** has great potential in the treatment of schizophrenia as a novel multi-target antipsychotic drug.

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
