# Peer review of "Design, Synthesis and Biological Investigation of Flavone Derivatives as Potential Multi-Receptor Atypical Antipsychotics"

_molecules, 2020, doi:10.3390/molecules25184107_

Round 1
Reviewer 1 Report
The manuscript is a continuation of previous authors` researches in the development of multi-receptor antipsychotic agents. This study describes the design, synthesis and evaluation of hybrid structures combining flavone-like and aryl piperazine/piperidine fragments as novel ligands of D2, 5-HT1A, 5-HT2A, and D3 receptors with improved pharmacological profile. A representative series of compounds was synthesized and the hit structure perspective for further optimization was found. The manuscript can be accepted after the following corrections:
1) On the p.2 (after fig.1) authors refer to their previous results, but the references [16,17] are not relevant.
2) The results of biological evaluation of compound 6j need to be moved in Conclusion, especially since the Introduction does not contain the structure of this compound.
3) In Scheme 1 the definition of “X” in formulas of arylamines should be included. In Scheme 1,3 (conditions iii) and Scheme 3 (conditions v) the general formula of the secondary arylamine should be included.
4) The authors do not use 13C NMR spectra to characterize compounds. At least, for the most active compounds, comprehensive data should be provided.
5) It would be useful to justify the trend in activity of the hit compound (6j) using computer-aided methods
6) In Supplementary the NMR spectra of compound 6j are unreadable.
Author Response
Re: Revised manuscriptMolecules-918316
Dear editor and reviewer,
The authors would like to thank the reviewers for their helpful comments which will improve the quality of the manuscript. We have carefully considered the comments provided by the reviewers and revised the manuscript accordingly. Enclosed please find the revised manuscript entitled “Design, synthesis, and biological investigation of flavone derivatives as potential multi-receptor atypical antipsychotics”, which we submit to for publication Molecules (Molecules-918316). Below you will find an itemized list of modifications made in accordance with the reviewer’s comments.
If you require any further information please do not hesitate to contact me. Thank you very much for your consideration. I am looking forward to hearing from you.
Best regards.
Guisen Zhang, Ph.D.
E-mail: gszhang@mail.hust.edu.cn
Tel.: +86-27-87792235
Fax: +86-27-87792170.
Reviewer 1
The manuscript is a continuation of previous authors` researches in the development of multi-receptor antipsychotic agents. This study describes the design, synthesis and evaluation of hybrid structures combining flavone-like and aryl piperazine/piperidine fragments as novel ligands of D2, 5-HT1A, 5-HT2A, and D3 receptors with improved pharmacological profile. A representative series of compounds was synthesized and the hit structure perspective for further optimization was found. The manuscript can be accepted after the following corrections:
- On the p.2 (after fig.1) authors refer to their previous results, but the references [16,17] are not relevant.
It has been revised according to the reviewer’s suggestion. (Line3 of Page3)
- The results of biological evaluation of compound 6j need to be moved in Conclusion, especially since the Introduction does not contain the structure of this compound.
We have already presented the structure of compound 6j in the abstract and graphic abstract. According to the reviewer’s suggestion, we added the chemical name of the compound 6j in the Introduction.(Line 13-14 of Page 3).Furthermore, we added some key data of the biological evaluation results of compound 6j in the Conclusion.(Line 8- 17 of Page 20)
The revised Conclusion were as below
Conclusion
In summary, a series of new flavone derivatives were synthesized based on heterocyclic and aromatic piperazine(piperidine), and their antipsychotic activities were evaluated. Of these compounds, 6j was favorable for binding to multiple acceptors with the KI value of D2 (12.1 nM), D3 (25.4 nM), 5-HT1A (9.7 nM) and 5-HT2A (3.2 nM) respectively. On the other hand, 6j showed low potency for 5-HT2C, α1, H1 receptors and human ether-a-go-go-related gene (hERG) channel (IC50 3769.8 nM), which were closely related to adverse effects of antipsychotics. In addition, compound 6j exhibited significant inhibition on schizophrenia-like symptoms (APO and MK-801-induced motor behavior) without causing catalepsy. Furthermore, compound 6j with its’ high value of ED50 (58.4mg/kg) in catalepsy and LD50 (>2000mg/kg) come to effect at a low dose in apomorphine-induced climbing test (ED50 0.19mg/kg), which displayed a wider range of the therapeutic indices compared with the atypical antipsychotics clozapine and risperidone. Thus, compound 6j has a great potential in the treatment of schizophrenia as a novel multi-target antipsychotic drug.
- In Scheme 1 the definition of “X” in formulas of arylamines should be included. In Scheme 1,3 (conditions iii) and Scheme 3 (conditions v) the general formula of the secondary arylamine should be included.
As the reviewer suggested, the definition of X” has been included in the scheme 1.At the same time ,the general formula of arylamines in Scheme 1 (conditions iii) and Scheme 3 (conditions v) were added in the description of scheme 1 and scheme 3 .
The revised Scheme1 were as below
The revised description of scheme1 and scheme 3 were as below
Scheme 1. Reagents and conditions: (i)(a) acetone, cyclopentanone or cyclohexanone, pyrrolidine, CH3CN, 50 ℃, 12h; (b) hydrochloric acid solution; (ii) 1-bromo-3-chloropropane or 1-bromo-4-chlorobutane, K2CO3, DMF, rt, 24h; (iii) with 4-aromatic ring substituted piperazine (6a-6i,6k,9a,9b,9e,9f,9i,9l) or 4-aromatic ring substitutedpiperidine(6j,6l,9c,9d,9g,9h,9k)CH3CN, K2CO3, KI, 70℃.(page 25 line 9 to 10)
Scheme 3. Reagents and conditions: (i) acetic anhydride, sodium acetate anhydrous, reflux, 14h; (ii) sodium bicarbonate solution, MeOH, rt, 3h; (iii)Pd/C, H2, rt; (ⅳ) 1-bromo-4-chlorobutane, K2CO3, DMF, rt, 24h; (ⅴ)6-fluoro-3-(piperidin-4-yl)-3a,7a-dihydrobenzo[d]isoxazole CH3CN, K2CO3, KI, 70℃.(page 27 line 5)
- The authors do not use 13C NMR spectra to characterize compounds. At least, for the most active compounds, comprehensive data should be provided.
The 13C NMR of compound 6j has been attached to the supplementary information.
- It would be useful to justify the trend in activity of the hit compound (6j) using computer-aided methods
Since the compounds we designed and synthesized mostly aimed at multiple targets (D2, D3, 5-HT1A 5-HT2A), we find it difficult to apply computer-aided method in this study. Thanks for the reviewer’s suggestion.In our future studies about single target drugs, we may introduce the computer-aided method.
- In Supplementary the NMR spectra of compound 6j are unreadable.
The 13C NMR and 1H NMR spectra of compound 6j were as below
13C NMR of compound 6j
1H NMR of compound 6j

Reviewer 2 Report
The authors submitted the MS for publication at the Journal. Studies In vitro and analysis of the Structure-Activity Relationship (SAR) results indicate that when 7-hydroxy-2,2-dimethylchroman-4-one group was selected as privileged structures, it derived from coumarin and flavone, the influence of various heterocyclic and aryl piperazines(piperidine) on the activities of D2, 5-HT1A and 5-HT2A receptors were evaluated and the results showed that when the phenylpiperazine was replaced by the phenylpiperazine with a benzoheterocycle-piperazines(piperidine), such as benzothiazole-piperazine and benzisothiazole-piperazine, given compounds 6h and 6i, their affinity for all the three receptors significantly increased. Especially, Compound 6j bearing 6-fluorobenzo[d]isoxazole-piperidine moiety displayed excellent D2 (Ki = 8.1 nM), 5-HT1A (Ki = 9.7 nM) and 5-HT2A (Ki = 3.2 nM) receptor potency.
Compound 6j stimulated the D2L, D3, 5-HT1A, 5-HT2A receptors in the agonist assay and showed weak agonist activity, the efficacy of the reference compounds less than 10%, respectively. In the antagonist assay, the efficacy of compound 6j blocked the four receptors more than 90%. Thus, 6j functioned as an antagonist at the D2L (IC50 = 8.9 nM), D3 (IC50 = 31.5 nM), 5-HT1A (IC50 =201.4 nM), 5-HT2A (IC50 =195.5 nM) receptors. Also showed low potency for 5-HT2c, α1, H1 receptors and human ether-a-go-go-related gene (hERG) channel, which closely related to adverse effects of antipsychotics. In addition, compound 6j exhibited significant inhibition on schizophrenia-like symptoms (APO and MK-801-induced motor behavior) without causing catalepsy. Furthermore, compound 6j displayed a wider range of the therapeutic indices compared with the atypical antipsychotics’ clozapine and risperidone. Thus, compound 6j has a great potential in the treatment of schizophrenia as a novel multi-target antipsychotic drug. Therefore, in a series of new flavone derivatives synthesized and evaluated their antipsychotic activities indicate that compound 6j have great potential …. Thus, I suggest accepting for publication at the Journal after minor revision. I suggest some minor changes as follow:
In Line 19 delete “Herein, we present the design and synthesis” and start The design of a …
In Line 52 after , add previously.
In Line 53 delete “in our previous studies”
Author Response
Re: Revised manuscriptMolecules-918316
Dear editor and reviewer,
The authors would like to thank the reviewers for their helpful comments which will improve the quality of the manuscript. We have carefully considered the comments provided by the reviewers and revised the manuscript accordingly. Enclosed please find the revised manuscript entitled “Design, synthesis, and biological investigation of flavone derivatives as potential multi-receptor atypical antipsychotics”, which we submit to for publication Molecules (Molecules-918316). Below you will find an itemized list of modifications made in accordance with the reviewer’s comments.
If you require any further information please do not hesitate to contact me. Thank you very much for your consideration. I am looking forward to hearing from you.
Best regards.
Guisen Zhang, Ph.D.
E-mail: gszhang@mail.hust.edu.cn
Tel.: +86-27-87792235
Fax: +86-27-87792170.
Reviewer 2
The authors submitted the MS for publication at the Journal. Studies In vitro and analysis of the Structure-Activity Relationship (SAR) results indicate that when 7-hydroxy-2,2-dimethylchroman-4-one group was selected as privileged structures, it derived from coumarin and flavone, the influence of various heterocyclic and aryl piperazines(piperidine) on the activities of D2, 5-HT1A and 5-HT2A receptors were evaluated and the results showed that when the phenylpiperazine was replaced by the phenylpiperazine with a benzoheterocycle-piperazines(piperidine), such as benzothiazole-piperazine and benzisothiazole-piperazine, given compounds 6h and 6i, their affinity for all the three receptors significantly increased. Especially, Compound 6j bearing 6-fluorobenzo[d]isoxazole-piperidine moiety displayed excellent D2 (Ki = 8.1 nM), 5-HT1A (Ki = 9.7 nM) and 5-HT2A (Ki = 3.2 nM) receptor potency.
Compound 6j stimulated the D2L, D3, 5-HT1A, 5-HT2A receptors in the agonist assay and showed weak agonist activity, the efficacy of the reference compounds less than 10%, respectively. In the antagonist assay, the efficacy of compound 6j blocked the four receptors more than 90%. Thus, 6j functioned as an antagonist at the D2L (IC50 = 8.9 nM), D3 (IC50 = 31.5 nM), 5-HT1A (IC50 =201.4 nM), 5-HT2A (IC50 =195.5 nM) receptors. Also showed low potency for 5-HT2c, α1, H1 receptors and human ether-a-go-go-related gene (hERG) channel, which closely related to adverse effects of antipsychotics. In addition, compound 6j exhibited significant inhibition on schizophrenia-like symptoms (APO and MK-801-induced motor behavior) without causing catalepsy. Furthermore, compound 6j displayed a wider range of the therapeutic indices compared with the atypical antipsychotics’ clozapine and risperidone. Thus, compound 6j has a great potential in the treatment of schizophrenia as a novel multi-target antipsychotic drug. Therefore, in a series of new flavone derivatives synthesized and evaluated their antipsychotic activities indicate that compound 6j have great potential …. Thus, I suggest accepting for publication at the Journal after minor revision. I suggest some minor changes as follow:
In Line 19 delete “Herein, we present the design and synthesis” and start The design of a …
Revised as the reviewer suggested (Line 18 of Page 1)
In Line 52 after, add previously.
In Line 53 delete “in our previous studies”
Revised as the reviewer suggested (Line 24 of Page 2)
